# Integrating Short Rotation Woody Crops into Conventional Agricultural Practices in the Southeastern United States: A Review

Omoyemeh J. Ile [1,*], Hanna McCormick [2], Sheila Skrabacz [2], Shamik Bhattacharya [2], Maricar Aguilos [1], Henrique D. R. Carvalho [3], Joshua Idassi [4], Justin Baker [1], Joshua L. Heitman [3] and John S. King [1]

1   Department of Forestry and Environmental Resources, NC State University, Raleigh, NC 27695, USA
2   College of Natural Resources, Environmental Science Program, NC State University, Raleigh, NC 27695, USA
3   Department of Crop and Soil Sciences, NC State University, Raleigh, NC 27695, USA
4   SC State 1890 Research & Extension, South Carolina State University, Orangeburg, SC 29117, USA
*   Correspondence: ojile@ncsu.edu

**Abstract:** One of the United Nations Sustainable Development Goal's (SDGs) aims is to enhance access to clean energy. In addition, other SDGs are directly related to the restoration of degraded soils to improve on-farm productivity and land management. Integrating Short Rotation Woody Crops (SRWC) for bioenergy into conventional agriculture provides opportunities for sustainable domestic energy production, rural economic development/diversification, and restoration of soil health and biodiversity. Extensive research efforts have been carried out on the development of SRWC for bioenergy, biofuels, and bioproducts. Recently, broader objectives that include multiple ecosystem services, such as carbon sequestration, and land mine reclamation are being explored. Yet, limited research is available on the benefits of establishing SRWC on degraded agricultural lands in the southeastern U.S. thereby contributing to environmental goals. This paper presents a literature review to (1) synthesize the patterns and trends in SWRC bioenergy production; (2) highlight the benefits of integrating short rotation woody crops into row crop agriculture; and (3) identify emerging technologies for efficiently managing the integrated system, while identifying research gaps. Our findings show that integrating SRWC into agricultural systems can potentially improve the climate of agricultural landscapes and enhance regional and national carbon stocks in terrestrial systems.

**Keywords:** land degradation; new farming systems; bioenergy; agroforestry; sustainable development goals; landscape restoration; ecosystem services

## 1. Introduction

The United Nations (UN) Sustainable Development Goals (SDGs) of restoring degraded lands, such as cultivated agricultural areas that have become eroded, abandoned, and unproductive, can help protect the environment and improve the productive capacity of ecosystems. In particular, the integration of tree species with annual agricultural crops has been identified as a restorative method for degraded agricultural lands while offering opportunities to produce fuelwood, pulpwood, bioenergy, and combined heat and power to generate renewable energy [1,2]. Woody biomass can supply bioenergy to supplement the increased use of renewable solar and wind energy [3] and create multiple job opportunities and energy security [4,5]. In the 2005 "Billion Ton Report", wood was identified as part of the bioenergy solution, where purpose-grown trees are expected to contribute significantly to account for 377 million dry tons of the 1.37 billion dry tons total biomass resource potential at projected yields of 17 Mg ha$^{-1}$ yr$^{-1}$. [6]. Therefore, short rotation woody crops (SRWC) that are coppiced (harvested) every three to five years can provide feedstocks for bioenergy and a regular source of revenues to farmers, while avoiding frequent soil disturbance and restoring soil health [3,7].

SRWC are fast-growing tree species grown as feedstocks for biomass, fuel, and fodder, the concept of which can be traced back to the early twentieth century by [8]. The research on SRWC began in the 1960s, where the concept of establishing sycamore trees (*Platanus occidentalis* spp.) in close spacing (1000–35,000 stems ha$^{-1}$) and short rotation cycles (1, 2, or 3 years) like agricultural row crops was developed in Georgia [9,10]. Following the Organization of the Petroleum Exporting Countries Oil Embargo of 1973–1974, a series of hardwood and softwood species had been investigated for bioenergy production, with objectives to evaluate productivity rates, rapid growth, and coppicing ability of hardwood species at different spacings and rotation lengths [11]. When SRWC are planted on degraded agricultural lands that are unproductive, and therefore not economically profitable, they reduce competition with food production that could occur with the expansion of biomass production [3,12]. Thus, restoring degraded soils to supply biomass for energy can ensure access to affordable, reliable, sustainable and modern energy for all (SDG Goal 7—Affordable and Clean Energy) [13]. Apart from restoring soil health such as improving the structure, microbial life, nutrient density, and carbon levels of the soil, SRWC can also restore lands polluted from fossil fuel extraction and contaminated with heavy metals [14–16].

The Southeast region has a rich agricultural and forestry history that is integral to the development of the United States (U.S.) economy [17]. The climate conditions of the region vary from warm to hot, and from dry to wet, with flat land, rich soil, and a long growing season that allow farmers in the region to grow crops for most of the year. Ten southeastern states (Alabama (AL), Arizona (AR), Florida (FL), Georgia (GA), Louisiana (LA), Mississippi (MS), North Carolina (NC), South Carolina (SC), Tennessee (TN), and Virginia (VA)) made up 21% of the 2017 U.S. net farm income, with notable cash crops such as cotton, peanuts, soybeans, corn, sugarcane, and tobacco [18]. In 2018, North Carolina led the Southeast in net farm income at over $4 billion [18]. The region collectively contributes nearly $47 billion to the nation's agricultural output each year, being the third-largest exporter of U.S. agricultural goods and top supplier of tobacco and livestock products [19]. For this reason, many Southeast farmers rely heavily on farm labor, accounting for nearly 20% of total farmworkers in the U.S. [19].

However, the southeastern (S.E.) U.S. is subject to extreme weather conditions, land area losses due to rising sea level, drought, hurricanes, [20–23] and climate change [24], which may be exacerbated by changes in land use [23]. The alternation of annual crops with SRWC in conventional farming systems could increase soil organic carbon (SOC), enhancing soil fertility, water retention capacity, and water infiltration rates, which in turn could increase productivity upon resumption of annual commodity crops [25,26]. Mann and Tolbert [27] explained that the tree canopy cover intercepts rainfall and decreases erosion potential, and fine root turnover improves nutrient cycling, reduces leaching losses, and increases organic matter inputs, while the forest floor intercepts surface runoff and enhances infiltration. Improved soil properties from establishing SRWC on eroded sites can be realized in as little as 3 to 5 years. These multipurpose restoration benefits serve as an effective climate change adaptation strategy that diversifies ecosystem services and income sources to landowners, while achieving the SDG goals 3, 13, and 15 of Good Health and Well-being, Climate Action, and Life on Land, respectively [28,29].

Integrating SRWC on marginal agricultural lands could also support regional economic development goals. Expanding wood pellet markets have led to regulatory policies related to investment, trade, and financial assistance that influence the forest biomass industry and promote biomass energy development [30–32]. In 2018, out of the five largest annual bioenergy electricity producing states, the Southeast recorded four states—Florida (5084 GWh), Georgia (4999 GWh), Virginia (4173 GWh), and Alabama (3446 GWh) due to its wide agricultural and forestry sectors, and California was number one (5946 GWh) [33,34]. Popular federal incentives in the S.E. region that encourage woody biomass production and utilization are United States Department of Agriculture (USDA) administered incentive programs such as the Rural Energy for America Program (grants and loan guarantees),

Community Wood Energy and Wood Innovation Program (directed at expanding wood energy markets and reducing wildfire risks), and Rural Energy Savings [34]. The most influential current federal policy that promotes the use of SRWCs for biomass is the production tax credit, while federal guidelines on the carbon emissions impact of woody biomass could also play a key role in guiding future investments [34]. Notwithstanding the financial incentives and potential income, some farmers are more influenced by non-financial factors such as identity, lifestyle, family, and farming culture. Therefore, biomass energy policy, especially regarding SRWC, should be tailored to the socio-cultural identity of proposed adopters [35].

Despite the multiple benefits of integrating SRWC into conventional agricultural systems and the potential of woody biomass expansion (Figure 1), the large-scale deployment of SRWC has been hindered by expensive bioenergy costs compared to fossil fuels and contrasting opinions on the impacts of bioenergy on sustainable development [36,37]. Burning of wood, the presumed adverse effects on biodiversity, and the relationship between land use change and bioenergy development are among the key issues debated [38–40]. Therefore, the purposes of this review are to document patterns and trends of integrated SRWC—conventional agricultural systems and provide comprehensive data on ecological, social, and economic benefits of the integrated system (Figure 1).

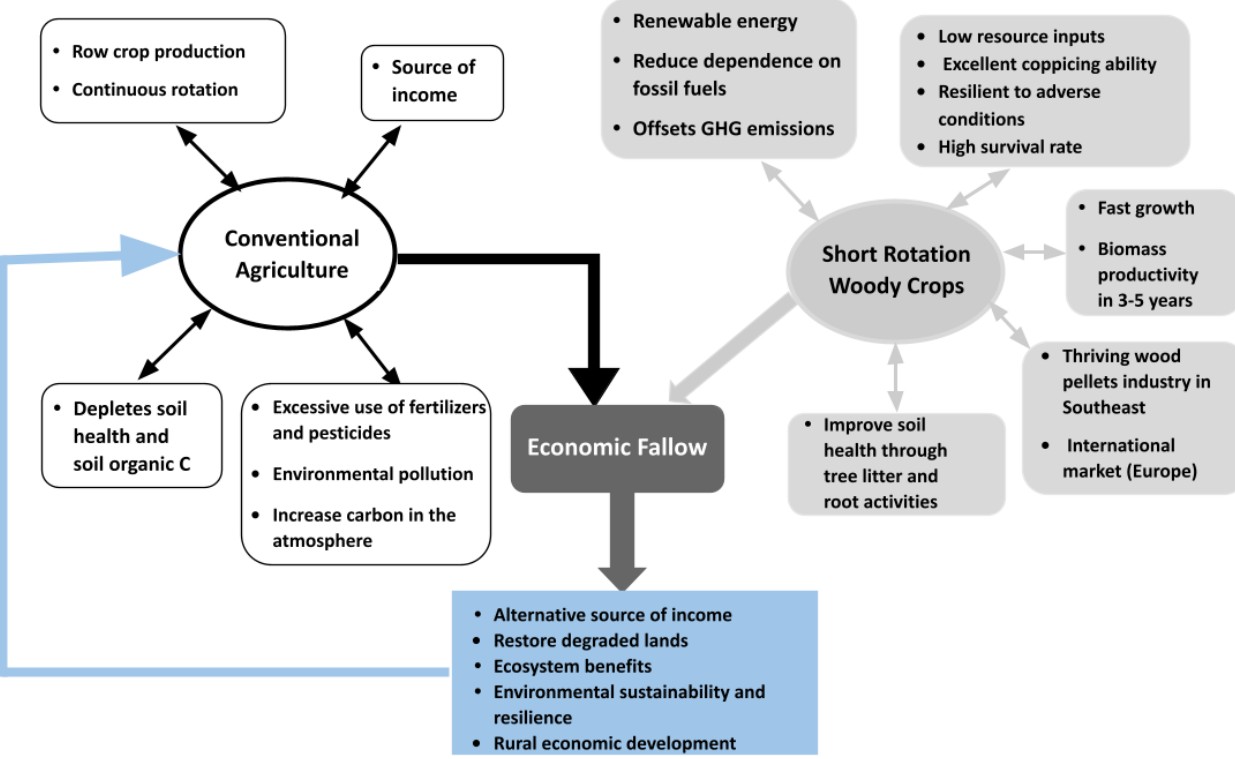

**Figure 1.** Conceptual model framework illustrating the attributes of conventional agriculture and short rotation woody crops and the potential benefits of an integrated system.

## 2. Methodology

We conducted a synthetic review on the integration of short rotation woody crops into agricultural systems in the S.E. through a general search of online scientific literature and major databases in Google Scholar, Scopus, CAB Directs, Web of Science, and AGRIS. To broaden our search, we also searched grey literature in Research Gate and university catalogs and reviewed extension publications, government documents, working papers, and conference proceedings. We used the following search strategies: short rotation woody crops in southeastern United States, agroforestry in southeastern United States, integrating bioenergy trees and row crops, integrating bioenergy trees and agriculture, emerging tech-

nologies in agroforestry, remote sensing in agroforestry, Geographic Information System (GIS) in agroforestry. We focused on the southeastern states as considered by the United States Geological Survey (AL, FL, GA, MS, NC, SC, TN, and VA). The search identified over 16,000 publications, which were narrowed down to 214 suitable publications using the following criteria: biomass and crop productivity, ecological benefits, economic benefits, social benefits, perceptions of stakeholders on integrated systems, applications of remote sensing in agroforestry. Due to the limited amount of research that has been done on integrating SRWC into conventional agriculture systems in the southeastern United States, we included studies from other regions of the country and Europe to buttress the discussion. We then used this information to present a detailed analysis of patterns and trends in SWRC bioenergy production, the benefits of integrating SRWC as purpose-grown feedstocks for bioenergy into row crop agriculture to restore degraded lands and identify emerging technologies for efficiently managing the integrated system. The analysis is divided into the following sections: (i) short rotation woody crops species in the region; (ii) ecosystem benefits of the integrated system; (iii) economic and social benefits of the ecosystem; and (iv) the role of remote sensing and GIS technologies in studying agroforestry systems. In regard to the latter (iv), although it is outside the scope of ecological, and socio-economic benefits of the integrated system, these technologies are critical to map suitable lands for establishing short rotation woody crops in the U.S. Southeast, and thus were an important consideration in our analysis.

## 3. Relevant SRWC Species in the Southeast Region

### 3.1. American Sycamore (Platanus occidentalis L.)

American sycamore trees are responsive to intensive silviculture and can be coppiced over several rotations [10,41]. They are tolerant of wet soils and are highly productive on sandy alluvial soils along streams and in bottomlands. Research has shown that American sycamore also thrives on degraded agricultural soils and is resilient to extreme environmental conditions, such as drought [42,43]. However, there has been limited research on the viability, productivity, and multi-use of American sycamore compared to the genera *Populus* L. in the S.E. Previous studies by the USDA Forest Service focused on its suitability for fiber production for the pulp and paper industry, and as a wood energy crop for co-fired power plants [44]. At establishment, American sycamore may be sensitive to water-availability [8,45] and to the endemic xylem disease bacterial leaf scorch, however, the coppice culture of sycamore trees keeps it in a juvenile state and reduces the likelihood of the trees being affected by diseases [46].

### 3.2. Eucalyptus (Eucalyptus spp.)

Eucalyptus species are highly productive on sandy clay loam and clay loam textured soils that are moderately to well-drained [47,48]. Growth is sensitive to water and nutrient availability and requires intensive weed control [49,50]. A possible issue with eucalyptus plantations is that eucalyptus could become an invasive species [51]. Another known issue with eucalyptus species is that they are intolerant to frost, reducing maximum photosynthesis rates and may cause severe stress when frosts happen repeatedly [52]. To combat this, there have been some clones developed with frost resistance, but they are still susceptible to freezing temperatures [51].

### 3.3. Loblolly Pine (Pinus taeda)

Loblolly pine has the fastest early growth of all southern pines [11]. It is drought-tolerant and can grow on soils ranging from very poorly drained to moderately well-drained [51,53]. Although, coppicing is not a feasible option for loblolly pine as compared to hardwoods, the pine may provide higher financial returns [51]. A study that varied irrigation and fertilization on loblolly pine showed no response to irrigation, indicating fertilization as a primary limiting factor for productivity [53,54]. While published literature on loblolly pine for production as a short rotation woody crop is not substantial, Kline

and Coleman, 2010 [51] suggested that loblolly pine was the best-adapted species for biomass in the S.E. region, topping hardwoods [51]. Loblolly pine is susceptible to a variety of pests and diseases, which can be limited through thinning and harvesting before maturity [55–58].

### 3.4. Poplar Species (Poplar spp.)

Eastern cottonwood trees show promise as they are the fastest growing poplar in North America, with high growth rates that slow after 4–8 years of tree development [51]. *Poplar* spp. can be easily propagated and thrive on well to moderately well-drained soils [59,60]. In natural conditions, trees achieve heights of 53–59 m with diameters of 120–180 cm, making them one of the tallest hardwood tree species [61,62]. Site-adapted genotypes and identified clones can be productive on marginal sites, up to 10.0 dry Mg ha$^{-1}$ yr$^{-1}$ and 13.0 dry Mg ha$^{-1}$ yr$^{-1}$ [59,63]. Intense management of competing species is highly important in the beginning of the rotation process as competition with weeds and other species can cause poor growth and establishment, especially in eastern cottonwood [64].

### 3.5. Sweetgum (Liquidambar styraciflua)

Sweetgum is tolerant to a large variety of conditions in the Southeast and was suggested to be more drought-tolerant than the other hardwoods based on irrigation studies [45,65]. The favorable biology of sweetgum allows it to be propagated from stem cuttings and coppiced over several rotations [53,66]. The intensive management and genetic improvement of sweetgum have increased productivity rates [66], however, because of its slow growth, sweetgum is not recommended for very short rotations [51].

## 4. Ecosystem Services and Environmental Sustainability

### 4.1. Biomass Productivity and Carbon Sequestration

#### 4.1.1. Biomass Productivity

SRWC grown for bioenergy are harvested on 3–10-year cycles, and in some cases, have been shown to persist for 20–30 years before yields start to decline [67]. The coppicing ability allows farmers to manage the tree plantations without replanting efforts. Typically, SRWC are harvested at about 10 cm above the ground during the winter season when the trees are dormant, after a significant amount of carbohydrates and nutrients have been translocated to the root system. Because of that, a large proportion of nutrients remains on site after harvest, contributing to SOC sequestration, and promoting future tree growth [11]. For optimum productivity, the trees are commonly planted at high densities, up to 12,000 to 14,000 trees ha$^{-1}$, at a spacing that allows agricultural machinery to work across the field [67,68]. Some research has shown that planting American sycamore at 5000 trees ha$^{-1}$ achieves the same productivity as 10,000 trees ha$^{-1}$ after the second rotation in the Piedmont region of North Carolina [46,69], suggesting a means to decrease establishment costs.

The growth and biomass productivity of SRWC is significantly dependent on genotype (species, family, clone), environmental conditions (climate, site quality, soil properties), and management [70,71] (Table 1). In the coastal plain of North Carolina, 89 poplar clones were tested for site suitability and biomass productivity [72]. Results showed 50–100% survival rate across clones, productivity ranged from 141 to 170 Mg ha$^{-1}$ for lower performing clones, 177 to 202 Mg ha$^{-1}$ for intermediate performing clones, and up to 215 to 226 Mg ha$^{-1}$ for high performing clones after an 8-year rotation [72]. Demonstrating the importance of site factors and physiographic regions, higher biomass productivity of 14.3 Mg ha$^{-1}$ yr$^{-1}$ for American sycamore in the coastal region of North Carolina at the end of the third rotation was reported, compared to 7.2 Mg ha$^{-1}$ yr$^{-1}$ in the Piedmont region [68,69].

The production of wood pellets for bioenergy in the southeastern United States allows for more efficient use of forest resources and improved forest management that encourages economic growth without environmental degradation. This is evidenced in the feedstock collected during forest harvest, promoting sustainable management, and the growth of sustainable green economy jobs. In addition, woody biomass provides affordable and

clean energy; hence, family woodland owners can play a significant role in providing feedstock for the wood pellet industry in the region, thereby contributing to the reduction of $CO_2$ and greenhouse gas (GHG) emissions. These strategies have positive environmental, social and economic impacts and can support multiple SDGs. Such as the SDGs 7, 8, 9 and 15—Affordable and Clean Energy; Decent Work and Economic Growth; Industry, Innovation, and Infrastructure; and Life on Land, respectively.

**Table 1.** Annual biomass productivity rates of candidate SRWC species in the Southeast.

| Species | State | Soil Properties | Rotation Age (Years) | Annual Biomass Productivity (Mg ha$^{-1}$ yr$^{-1}$) | Inputs | Reference |
|---|---|---|---|---|---|---|
| American Sycamore | NC | Sandy loam | 3 | 11.1 | NA | [46] |
| | NC | Sandy loam | 5 | 9.0 | NA | [46] |
| | NC | Loamy fine sand | 3 | 22.3 | H, IN | [69] |
| | SC | Sandy | 11 | 8.7 | NA | [53] |
| | FL | Sandy loam | 8 | 3.4 | NA | [73] |
| | FL | Sandy loam | 8 | 6.3 | IRR | [73] |
| | SC | Sandy loam | 5 | 5.3 | NA | [66] |
| | SC | Sandy loam | 7 | 8.0 | NA | [66] |
| | AL | Sandy loam | 4 | 7.3 | NA | [47] |
| *Eucalyptus* spp. | FL | NA | 7 | 42.0 | NA | [74] |
| | SC | NA | 6 | 34.8 | NA | [75] |
| | NC, SC, FL, GA, AL | NA | 6–10 | 13.7–26.4 | NA | [48] |
| Loblolly Pine | SC | Sandy | 11 | 13.8 | NA | [53] |
| | SC | Sandy | 11 | 21.4 | IRR, F | [53] |
| | GA | NA | 6 | 15.2 | 1RR, F, P, H | [76] |
| | NC | Sandy loam | 6 | 12.4 | NA | [43] |
| | FL | Sandy | 13 | 6.8 | F | [77] |
| | FL | Sandy | 13 | 5.9 | H | [77] |
| | GA | Sandy | 4 | 21.4 | H, F | [78] |
| *Poplar* spp. | NC | Loamy Sand | 10 | 5.3 | H | [43] |
| | NC | Loamy fine sand | 6 | 3.6–18 | NA | [79] |
| | GA | Silty clay | 3 | 2.4–4.2 | IRR | [80] |
| | FL | Sandy loam | 8 | 6.6 | IRR | [73] |
| | SC | Sandy | 3 | 3.1 | IRR, F | [45] |
| | SC | Sandy | 3 | 0.6 | NA | [45] |
| Sweetgum | SC | Sandy | 11 | 21.3 | IRR, F | [53] |
| | SC | Sandy loam | 7 | 17.5 | H, F | [66] |
| | GA | NA | 6 | 10.9 | IRR, P, F | [76] |
| | GA | NA | 6 | 2.1 | NA | [76] |
| | MS | Silt loam | 4 | 6.7 | NA | [81] |
| | NC | Sandy loam | 6 | 12.1 | IRR, H | [43] |

Rotation age in years = Biomass productivity after "n" years. Inputs: F = fertilizer, H = herbicide, IN = insecticide, IRR = irrigation, NA = not available, P = pesticide.

4.1.2. Aboveground Carbon Sequestration

Short rotation woody crops offer an alternative to fossil fuels, and they could reduce the rate of enrichment of atmospheric $CO_2$ due to high biomass accumulation, providing an opportunity to mitigate the effects of GHG emissions and resulting global change [79]. A report from USDA and U.S. Department of Energy projected that annually about 342 million dry tons of biomass could be derived from bioenergy crops [6]. Therefore, interest in carbon sequestration has played a prominent role in the rotation of trees with row crops in regions of extensive agricultural land use [6,11].

Cook and Beyea [82] estimated biomass productivity of 7.4 and 8.0 Mg C ha$^{-1}$ yr$^{-1}$ of SRWC willow and poplar in a 3-year and 10-year rotation, respectively, compared to 5.4 Mg C ha$^{-1}$ yr$^{-1}$ of corn. They explained the SRWC cropping systems could sequester

up to 500 kg C Mg$^{-1}$ (kg of C per Mg of biomass) in willow, and 600 kg C Mg$^{-1}$ in poplar, compared to only 300 kg C Mg$^{-1}$ in corn biomass. At the end of a 5-year rotation of Poplar SRWC, between 9.1 and 10.7 Mg ha$^{-1}$ of biomass was estimated to remain on site of the converted cropland following harvest, largely from input of the stumps and perennial root systems [83]. As a result, extensive implementation of establishing SRWC on degraded agricultural landscapes is underway to produce verifiable agroforestry C offsets and renewable biomass feedstock supplies [84]. This is important for the southeastern U.S. region, where intensive agriculture has been practiced for decades and plantation forestry is economical [85].

### 4.1.3. Belowground Carbon Sequestration

Planting SRWC on degraded and eroded agricultural lands left to fallow has been shown to enhance C sequestration rates from 0.6 to 3.0 Mg C ha$^{-1}$ yr$^{-1}$ [84]. An important component of SRWC for soil carbon sequestration is prolific root systems and fine root turnover, enhancing soil organic matter and the microbial community, as well as the capacity to retain/supply water and nutrients to plants [86,87]. The composition and activity of the microbial community is a major driver of soil carbon cycling, regulating the turnover of nutrients and decomposition of soil organic matter, which are important to enhance long-term soil carbon storage as well as sustainable agricultural soil management [86–88].

A global meta-analysis of soil carbon sequestration by Qin et al. [89] reported that cropping of marginal agricultural lands with SRWC provides an opportunity to sequester 0.3–1.7 Mg C ha$^{-1}$ yr$^{-1}$. In Europe, Georgiadis et al. [90] quantified the effects of converting agricultural fields to SRWC poplar and willow, where the average C concentration ranged from 11 to 68 Mg C ha$^{-1}$ in agricultural fields, and from 15 to 82 Mg C ha$^{-1}$ in SRWCs. SOC also increased after 4 years of converting agricultural lands to SRWC poplar from 10.9 to 13.9 kg C m$^{-2}$ [87]. In the U.S. Southeast, SRWC plantations established on former agricultural fields sequestered soil C at rates ranging from 40 to 150 g C m$^{-2}$ yr$^{-1}$ to a depth of 40 cm in the soil (Table 2).

**Table 2.** Soil carbon inventory of SRWCs plantation on former agricultural fields (control site).

| SRWCs | Control Site | State | Soil Texture | Rotation Age (Years) | Soil Carbon Stock (g C m$^{-2}$) (SRWC Site) | Soil Carbon Stock (g C m$^{-2}$) (Control Site) | Sequestration Rate (g C m$^{-2}$ yr$^{-1}$) |
|---|---|---|---|---|---|---|---|
| Loblolly Pine | Old Ag. Field (Annuals and perennial grasses) | South Carolina | Sandy Clay | 6 | 3505 ± 170 | 2595 ± 110 | 152 |
| Loblolly Pine | Fallow Ag. Field | South Carolina | Sandy Clay Loam | 11 | 2956 ± 82 | 2483 ± 144 | 47 |
| Sweetgum | Old Ag. Field (Annuals and perennial grasses) | Tennessee | Loam | 11 | 4649 ± 289 | 3676 ± 82 | 88 |
| Sycamore | Old Ag. Field (Annuals and perennial grasses) | Tennessee | Loam | 11 | 4100 ± 239 | 3676 ± 82 | 39 |

Plots were marked for control measurements (control site) adjacent to SRWCs plantation. Carbon accumulation was estimated as the difference between soil C stocks under tree plantations and reference sites. Annual rate of soil C sequestration was estimated under SRWC plantations. Data adapted from Charles et al. [91].

Furthermore, studies have reported increased SOC after cropland conversion to SRWC in the upper 30 cm soil layer compared to deeper layers [92–94], and use of no-till systems in the management of SRWC may lead to an accumulation of surface residue carbon and nitrogen [95]. Therefore, some studies highlight the importance of sampling deeper layers, particularly in long-term studies [96,97]. Kahle et al. [98], concluded that carbon storage and sequestration under SRWC plantations was significantly higher compared to crops. Hence, it is imperative to assess region-specific soil carbon cycling and sequestration in degraded agricultural lands converted to SRWC and the effects on ecosystem processes.

The active restoration of degraded agricultural lands with SRWC to enhance above and below ground carbon storage and sequestration can substantially contribute to SDG 13 (Climate Action) by strengthening resilience and adaptive capacity to climate change. Further, strategies implemented to achieve SDG 13 would have significant effects on the pursuit of all other SDGs, benefiting goals relating to food, energy, and water availability, community development, poverty eradication and employment, justice and equality, and sustainable ecosystems. This is because the UN SDGs reinforce their commitments to each other. For example, the approach of SWRC afforestation ensures the progress of Climate Action—SDG 13 and SDG 7—Affordable and Clean Energy while contributing to Clean Water and Sanitation—SDG 6, Responsible Consumption and Production—SDG 12, and Life on Land—SDG 15.

### 4.2. Short Rotation Woody Crops for Regenerative Agriculture

The integration of SRWC on eroded agricultural fields can improve soil physical properties [99] and enhance environmental benefits associated with soil biological processes carried out by soil microorganisms [100,101]. There is more microbial diversity in agroforestry systems than in conventional agricultural cropping systems as a result of tree litter and tree root exudates contributing to the improvement in soil properties [102–104]. For example, larger water-stable soil aggregates were recorded in undisturbed soils of a forest plantation compared to smaller aggregates in cultivated soils [105]. Similarly, Devine et al. [106] observed that the conversion of a soybean field to American sycamore plantation increased the mean diameter of aggregates compared with row crops, from 0.96 mm to 1.30 mm after 5 years of management. Increased soil aggregate stability in SRWC has been attributed to the presence of decomposed organic material that binds soil particles into stable aggregates, thereby improving soil structural properties [107].

Kahle et al. [108] reported that six years after converting cropland to *Salix* and *Populus* SRWC, soil porosity increased, and soil bulk density decreased. Similarly, the bulk density of a cropland topsoil that was 1.65 g cm$^{-3}$ decreased to 1.35 g cm$^{-3}$ after 18 years of establishment of SRWC [109]. Furthermore, soils under SRWC have demonstrated higher cumulative water infiltration rates and hydraulic conductivity than soils under row crops due to higher macro—porosity under the trees [110]. Likewise, there was an increase in plant available water under SRWC in the upper soil layer (0–10 cm depth) compared to cropland [109]. The researchers attributed the increase to high soil organic matter and water-stable soil aggregates that were also found in the 0–10 cm soil depth, similar to results of Kahle et al. [98]. Depending on the planting density, SRWC have been shown to improve soil water retention and air-filled pore space of the soil [99], and other studies have linked productivity and yield of bioenergy establishments to available soil water content [111,112].

Various strategies have been proposed to mitigate N losses from row crop agriculture such as crop rotation, cover cropping, crop residue, appropriate N fertilization rates, use of nitrification inhibitors, and water management in irrigated crops [113–115]. The integration of SRWC on eroded agricultural fields also has potential to reduce site N losses. Deep tree roots can take up nitrogen that leaches below the agricultural crop root zone [116]. In a conversion of row crop agriculture to trees, the tree establishment facilitated the reduction of N losses via nitrate ($NO_3^-$) leaching and soil nitrous oxide ($N_2O$) emissions during the establishment years [99]. By the third year of tree establishment, $NO_3^-$ leaching was significantly lower, at 2.7 kg N ha$^{-1}$ yr$^{-1}$, compared to the corn-soybean field, at 3.9 kg N ha$^{-1}$ yr$^{-1}$ [117]. Furthermore, tree roots have the capacity to reduce N loss in deeper soil layers depending on the species and soil texture [118,119].

Short rotation woody crops require less disturbance and minimal inputs during establishment and maintenance compared to annual crops, therefore, they are expected to reduce soil erosion and improve soil health [120]. SRWC have been shown to reduce sediments, pesticides, and nutrients losses to surface runoff and groundwater due to extensive canopy cover and root systems, thereby improving water quality [121,122]. This is also facilitated by the interception of rainfall by tree canopy cover, the infiltration of precipitation, and

the amount and timing of evapotranspiration, all of which are important for soil physical, chemical, and biological properties [122,123]. In Mississippi, there was a higher amount of sediment loss, at 16.2 Mg ha$^{-1}$, in a tilled cotton (*Gossypium hirsutum* L.) field, compared to 2.3 Mg ha$^{-1}$ for eastern cottonwood (*Populus deltoides*) over 14 months [121]. Likewise, in Tennessee, sediment loss was significantly higher under no-till corn (*Zea maize* L.) than sycamore (*Platinus occidentalis* L) and significant amounts of $NO_3^-$ were detected in the runoff water from the corn plots [121]. Researchers explained that over 80% of the sediment lost in the cotton field happened in one rainfall event (about 57 mm rainfall), ascribing the massive loss to soil crusting and decreased infiltration rates [121].

Another benefit of SRWC in conventional agriculture is weed suppression. In the coastal plain of NC, where an American sycamore SRWC plantation was converted back into a corn field, we observed a difference in weed quantity in the alleys (between tree plots) vs. inside the former tree plots (Figure 2). Although weed seed banks were not quantified, the report of Fischer et al. [69] showed a higher reduction of weed biomass in sycamore plots compared to other SRWC species, supporting our observation. The cumulative beneficial effects of tree rooting, tree litter, accumulated organic matter, reduced soil tillage, and absence of pesticides of establishing SRWC on degraded agricultural fields moves us closer to the regenerative approach, resilience to climate change, and the strengthening of healthy soils.

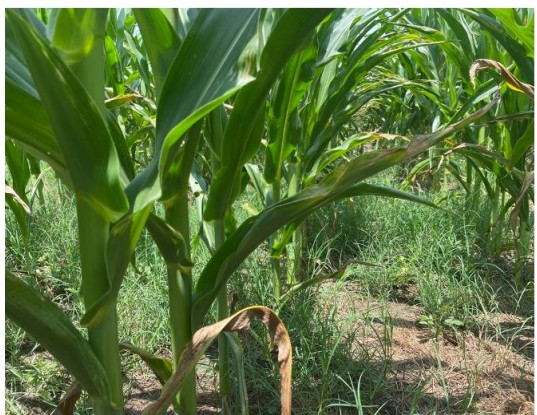 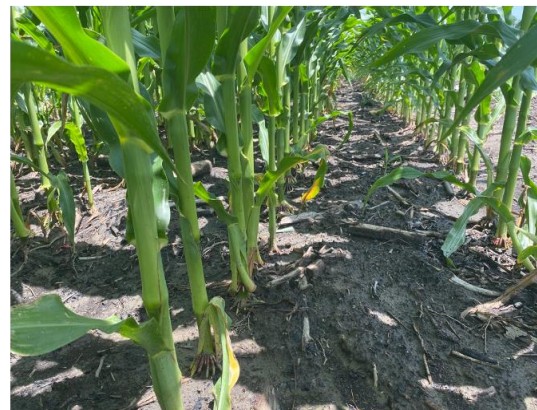

A. Corn in alley between tree plots          B. Corn in tree plots converted into ag. field

**Figure 2.** Left—(**A**. Corn in alley between tree plots): more weeds in corn rows planted in alleys between former tree plots. Right—(**B**. Corn in tree plots converted into ag. field): reduced weeds and tree litter in corn rows planted in former sycamore tree plots. Pictures were taken by the first author one month after corn was planted at Wallace, Duplin County, NC.

The SDG to end hunger and achieve food security (SDG 2) recognizes the inter linkages among supporting regenerative agriculture, land, healthy soils, and sustainable production and consumption food systems. Therefore, integrated decision-making processes are needed to adequately address these trade-offs to increase resource-use efficiency, (SDG 12—Responsible Consumption and Production) and the transition towards low-carbon and green economies (SDG 9—Industry, Innovation, and Infrastructure and SDG 13—Climate Action).

### 4.3. Increased Biodiversity

In a study on the environmental effects of SRWC for bioenergy in the S.E. region, Griffiths et al. [124] explained that SRWC systems could influence ecosystem communities. Correspondingly, researchers that reviewed the impacts of SRWC poplar on bird and mammal diversity in the U.S. found higher species diversity in SRWC compared to agricultural lands, but lower species richness than reference unmanaged forests [124,125]. A study conducted in Florida, Georgia, and Alabama indicated only one species of conservation

concern occurred in a cornfield while all seven species of concern occurred widely across SRWC plantations [126]. Likewise, other studies have indicated that SRWC pine supported more avian diversity than maize fields, but less than when compared to unmanaged forests [127,128]. This has been attributed to the canopy cover of pine forests being a major driver of biological diversity in the S.E. [129].

A review of the literature on animal diversity in SRWC found that willows generally showed greater biodiversity and abundance than poplars, but the results varied by clones, where birds favored certain willow clones as nesting sites [130]. Although wildlife species diversity may be lower in SRWC stands relative to unmanaged forests, Riffell et al. 2011 [125] hypothesized that the differences might decrease with increasing age of the SRWC stands. In contrast, Hamel [131] found more avian diversity in 4- to 6-year-old eastern cottonwood stands than in stands of slower-growing hardwoods. The author attributed the fast growth and canopy cover of the eastern cottonwood stands in encouraging an assembly of the avian community compared to stands of slower-growing trees [131]. Integrating fast growing hardwood tree SRWC into agriculture could provide an opportunity for birds to serve as an effective biological control agent to reduce crop damage and pest abundance [132–134].

Though the possibility of harvesting SRWC after a short rotation (3–5 years) may affect the avian and mammalian diversity, it could contribute to overall landscape biodiversity. At a landscape scale, if the establishment of SRWC leads to reduced harvesting of natural forests, this may increase the protection of natural forests as well as the avian and mammalian diversity. However, more research is required on how the age, height, and heterogeneity of SRWC influence stand level and landscape biodiversity. The impact of increased biodiversity as a result of integrating SRWCs into conventional agricultural practices to restore degraded lands is a strategy towards achieving the SDG 15—to protect and sustainably manage forests, to reverse land degradation, and to halt biodiversity loss.

## 5. Economic and Social Factors

### 5.1. Return on Investment and Financial Implications

Forestry and agriculture are important parts of the economy for many states in the southeast U.S., hence integrating SRWCs on marginal agricultural lands can produce a non-tax-funded economic return to the farmer [135]. It has been reported that farmers are motivated by crops that deliver direct economic return as well as environmental benefits [136,137]. Profitability for these crops can come from the rapidly growing wood pellet global market and the rising demand for energy that is low-carbon, efficient, and environmentally benign [34,138,139]. Further, SRWCs offer a potential source of revenue in multiple potential product pools; SRWC poplar have been tested for engineered or panel products such as laminated veneer, if proven profitable, it could expand the socio-economic benefits of SRWCs [72,140].

Integrating SRWC into agriculture can also financially benefit farmers in indirect ways. Farmers could see lower production costs overall through reduced pesticide and nutrient and the related outcome of environmental pollution. [36,132,134]. Furthermore, landowners vested in non-industrial forest products, such as aesthetics and wildlife, can increase economic benefits by attracting birds and other wildlife to their farms at higher rates than under baseline production technologies [140]. With SRWC implemented as windbreaks, it will take a shorter amount of time for the windbreak to go into effect, increase crop yields, and possibly offset the costs of the windbreaks themselves through the revenue potential from final harvests [141–144]. Other integrated SRWC systems include riparian buffers that could improve financial returns, as they have been shown to be cost-beneficial with bioenergy crops [145].

Studies have reported low crop yield when trees are integrated into row crop production via alley cropping, depending on cropping patterns and other practices. Cubbage et al. [146] reported low crop yields and negative financial returns in an alley cropping system of loblolly pine and corn-soybean row crops in the coastal region of North Carolina.

This is also supported by modeling that has shown mono-cropping provides higher profits than alley cropping, this may be as a result of competition for resources (e.g., solar radiation, water, nutrients, etc.) between trees and row crops [147–150]. Nonetheless, the integrated system could provide modest financial returns from the trees much more than only the row cropping system could on a poor site and in adverse weather conditions [146]. In addition, growing SRWC has been estimated to increase the net present value of the agriculture and forest sectors by about $6 billion, thereby representing a substantial share of woody biomass supply in the future [140].

Comparing the woody species typically grown for biomass in the Southeast, loblolly pine has provided the highest financial return [151]. A study on the financial returns of short rotation loblolly pine biomass in the piedmont region of Virginia and the coastal plain of North Carolina, reported an internal rate of return of 8.3–9.9% using the 2019 pulpwood price [151]. Using a 5% discount rate, the authors reported the break-even stumpage price was $8.72–$9.92 $Mg^{-1}$ and the break-even yield was approximately 18 $Mg\ ha^{-1}\ yr^{-1}$ productivity [152]. Another study in the South reported a positive Net Present Value at 5% discount rate, a $10 $Mg^{-1}$ stumpage for 12-year loblolly pine biomass plantations [152]. Lower break-even-prices at 5% discount rate have been recorded for well-drained upland sites versus poorly drained lower sites due to lower cost of site preparation and management [153,154]. SRWC biomass plantations could be more cost effective and yield higher financial returns when site establishment, resource inputs, transportation of feedstock to processing mills, and general maintenance are minimized while pulp prices increase [155].

*5.2. Potential Revenue from Carbon Markets and Other Incentive Programs*

Emerging policy priorities at the U.S. federal and state levels to increase land carbon sinks and expand renewable energy production could support a growing market for SRWCs. For instance, an expanded domestic (or global) market for wood-based bioenergy could stimulate investment in forestry or integrated agroforestry, including fast-growing plantation systems like SRWC, while supporting carbon sequestration goals [156,157]. New programmatic efforts in the U.S., including the Partnerships for Climate-Smart Commodities, could support further expansion of SRWC and integrated systems given their relative climate benefits compared to traditional farming practices and flexibility in providing biomass to multiple potential product pools (agricultural commodities, pulpwood, and bioenergy).

Another area of potential future revenue for integrated SRWC systems could be voluntary carbon markets, which are emerging as a potential source of revenue for farmers that adopt practices that increase carbon sequestration rates above- and belowground (compare to business as usual) are known as carbon farming [156,158]. This type of "carbon farming" relative to some lower sequestration baseline could generate carbon credits that could be rewarded in voluntary markets. In the case of integrated SRWC systems, farmers could potentially claim carbon benefits in both trees and soils. While no protocols currently exist for SRWC systems, one could envision a methodology being developed in the near future given expanding private sector interest and financing around carbon offset markets through, the California Air Resources Board (CARB) carbon market and other voluntary offset markets.

Agroforestry and forestry ecosystems are efficient in carbon farming, and as such, they have been given priority in carbon trading as an instrument of conservation, as well as the ability to reduce GHG emissions and store carbon [159–162]. Consequently, integrating SRWC into agriculture provides an opportunity for farmers and landowners in the southeastern U.S. to engage in the growing carbon offset market and forest carbon credit programs, at a rate of $12.00 to $18.00 per ton of carbon sequestered [163]. Projections estimate that by the year 2030, the C offset market could range from $5 billion to $30 billion at the low end to more than $50 billion at the high end [164]. However, there have been contrasting views on the cost-effectiveness of carbon sequestration from forest and agricultural activities. Some authors suggest that forest-based carbon sequestration is the most cost-

effective [165,166], whereas others prefer an increased demand for bioenergy feedstocks and increased net farm income from afforestation/SRWC [159], while still others argue that agricultural activities will be most cost effective [167]. Recent estimates from an integrated multi-sector modeling framework indicate that mitigation from forest management and afforestation would capture more than two-thirds of the potential abatement from the U.S. land use systems under comprehensive climate policy scenarios [168].

Despite these arguments, concerns on the longevity of SRWC carbon sequestration potential remain. First, farmers could face near-term revenue losses post-SRWC establishment on more productive agricultural lands. In such contexts, additional compensation (via carbon markets or public incentive program like the conservation reserve program) could be necessary to help farmers avoid lost revenue from traditional crop rotations while the SRWC system is established. Further, regional analysis of the costs of biomass transport, storage and processing is needed to assess market potential for the SRWC. Regional analyses that link farm-scale economics with regional distribution and processing networks can be used to quantify the additional incentives needed to ensure revenue parity between traditional commodity production and SRWC (see Masum et al. [169] for a related example of carinata production systems in the U.S. South).

Further, potential carbon leakage may occur from conventional agricultural practices on the remaining agricultural land or the conversion of SRWC back to agricultural uses may reduce the projected quantity of carbon sequestered [162,170]. In addition, the requirements for landowners through the CARB carbon market are more stringent, like a one-hundred-year conservation easement, carbon sequestration verification every six years, and a large land acreage (800–1200 hectares) could be required to generate a positive net profit [163]. The voluntary carbon market offers a flexible program for landowners to enroll smaller land acreage for a shorter time period, as short as 1 year. Therefore, the higher likelihood of payments to farmers who plant SRWC through the voluntary program could serve as an incentive to potential biomass producers and provide an alternative source of income to farmers while restoring degraded lands [171].

*5.3. Perceptions of Stakeholders*

Integrating short rotation woody crops on degraded agricultural lands left to fallow can sustain rural livelihoods through the growing wood pellet industries of the Southeast [172–174]. Though the integration may seem simple, the practice may be complex, ranging from the individuality of the farmers, motivations and influencers, inadequate information, new and uncertain terrain [175–178]. Farmers may find it difficult to design the specific integrated farming system to meet their diverse objectives and maximize the aforementioned benefits [175,178,179]. Therefore, an in-depth assessment and understanding of farmer perceptions and behaviors towards integrating SRWC on marginal lands is required [180–182]. These include the social and environmental motivations as well as behavioral approach theories to environmental sustainability [183,184]. Studies have reported significant correlations between these motivators and the decision to participate in SRWC/agroforestry programs among landowners [185–187].

Challenges that have been identified in the literature from the farmer's perspective include high investment costs, long-term encumbrance of land and capital, low prices for SRWC biomass, poor relation to farmer socio-cultural identity [187], while low resource inputs, income opportunity, and ecosystem benefits are reported as incentives [188,189]. Scholars have argued that it is challenging to decide on a general approach to enhance farmer willingness to adopt SRWC, because these factors differ case by case, region by region, and are based on individual farmer goals [190]. For example, some farmers believe that SRWC can improve crop yields, while others noticed environmental benefits such as improved soil fertility and increased biodiversity [191]. In a survey by Hodges et al., 2019a, 2019b [191,192] in the Chesapeake and Savannah fuel shed areas (top two US ports for wood pellets), a majority of landowners were found to be neutral regarding their understanding of woody biomass energy (Figure 3a) however, they are more likely to grow and sell woody

biomass if they are provided with the right incentives, access to markets, and technical assistance (Figure 3b).

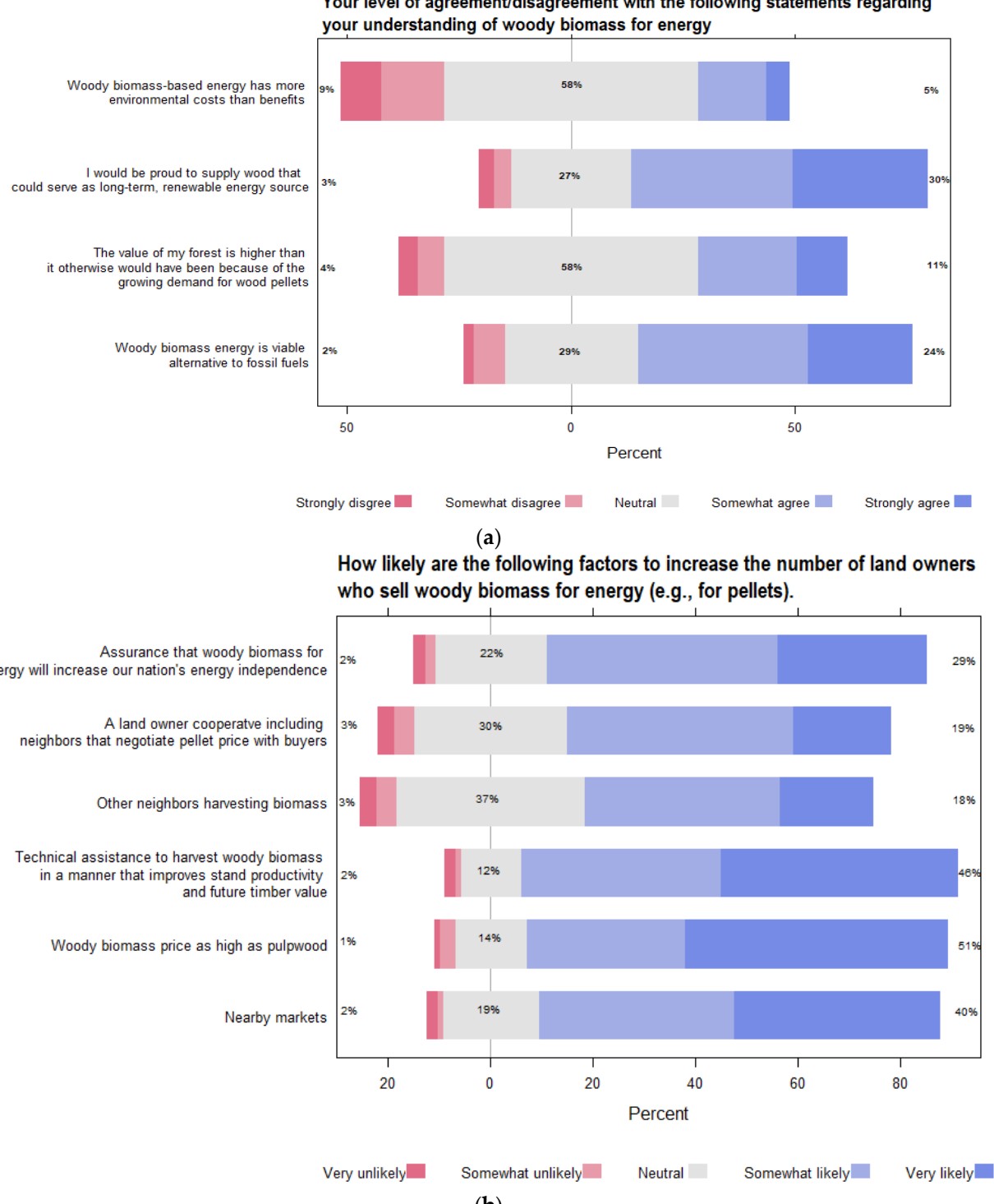

**Figure 3.** (**a**) Percentage of landowner's perceptions about woody biomass for energy. Percentages labelled on the graph are only for the strongly disagree, neutral, and strongly agree results. Data adapted from Hodges et al. [193]. (**b**) Percentage of landowner's perceptions on incentives to sell woody biomass. Percentages labelled on the graph are only for the strongly disagree, neutral, and strongly agree results. Data adapted from Hodges et al. [192].

Therefore, with adequate resources, such as extension training and outreach materials that will meet the specific needs of each farmer/landowner as well as field trials, farmers and landowners can gain the information and support they need to potentially adopt new integrated farming systems. The USDA Biomass Crop Assistance Program, which contractually connects biomass producers with end users, provides technical advice, cost share funding, and subsidized biomass prices as incentives to adopt SRWC [194]. However, it is important to note that their last fiscal year was in 2017. The perceived longer wait-times involved in investing in trees compared to annual row crops may serve as a barrier to farmers that could be addressed through subsidies, well-developed markets [195], and technology transfer that emphasizes staggered planting schedules for blocks of SWRC allowing for generation of annual revenues.

A case study on the adoption of SRWC among farmers in North Carolina using behavioral approaches highlighted that some farmers may be interested in growing short rotation woody crops if they can harvest the trees in three to five years [179]. The authors found that the landowners that are willing to diversify their farm economies with SRWC are multi-purpose landowners, or those that own land mainly for recreation/protection and family/personal uses. Likewise, Joshi and Mehmood [193] found that bioenergy conservationists, multiple-objective landowners, and passive landowners would be more willing to grow SRWC and supply wood-based biomass in Florida and Virginia. Studies have also listed market availability, education and awareness, funding, and farmer social networks as influences that could hinder or facilitate the adoption of SRWC [196–198]. Their recommendations were to target young farmers because they are more open to trying new techniques and farmers with low productive agriculture land close to wood pellet manufacturing facilities [180].

## 6. Emerging Technologies

Natural resources are easily affected by climatic variations and other biotic and abiotic stressors, hence the need for all these factors to be analyzed on a spatial-temporal basis for effective management and decision-making. In the restoration of degraded lands, emerging technologies can offer resources that enable data collection, processing, storage, and analysis across systems such as spatial prioritization techniques and automation of ecosystem monitoring to manage and mitigate environmental change [199–201]. In the past decades, advanced techniques like remote sensing, global positioning system, and GIS have been used to enhance the assessment, management, and monitoring of agriculture, forestry and related ecological processes [202]. Therefore, the use of remote sensing and GIS provide consistent, timely and valuable information for large-scale applications in agriculture, forest resource monitoring, and for policymakers and decision-makers [203].

When remotely sensed data are integrated with field data, and other geospatial information systems, they can be used by resource managers and researchers in monitoring of SRWC/agroforestry establishment programs, forest inventory-based biomass assessments, and land suitability assessments [204]. We analyzed the land suitability of SRWC in six southeastern states (Alabama, Florida, Georgia, North Carolina, South Carolina, Virginia) using three suitability criteria: (1) marginal land, (2) lands that have been intensively cropped with corn, soybeans, and tobacco, (3) non-federal lands. We used the USDA Soil Survey Geographic Database (SSURGO) and identified land capability classes 5–8 as marginal lands [205]. The SSURGO classifies U.S. lands into eight distinct categories using soil and biophysical environmental properties. As the number increases from 1 to 8, the land suitability for vegetation growth decreases. Results show that Florida has the largest amount of marginal land, at ~48,474,354 hectares, while Virginia has the least at ~115,585 hectares (Figure 4a).

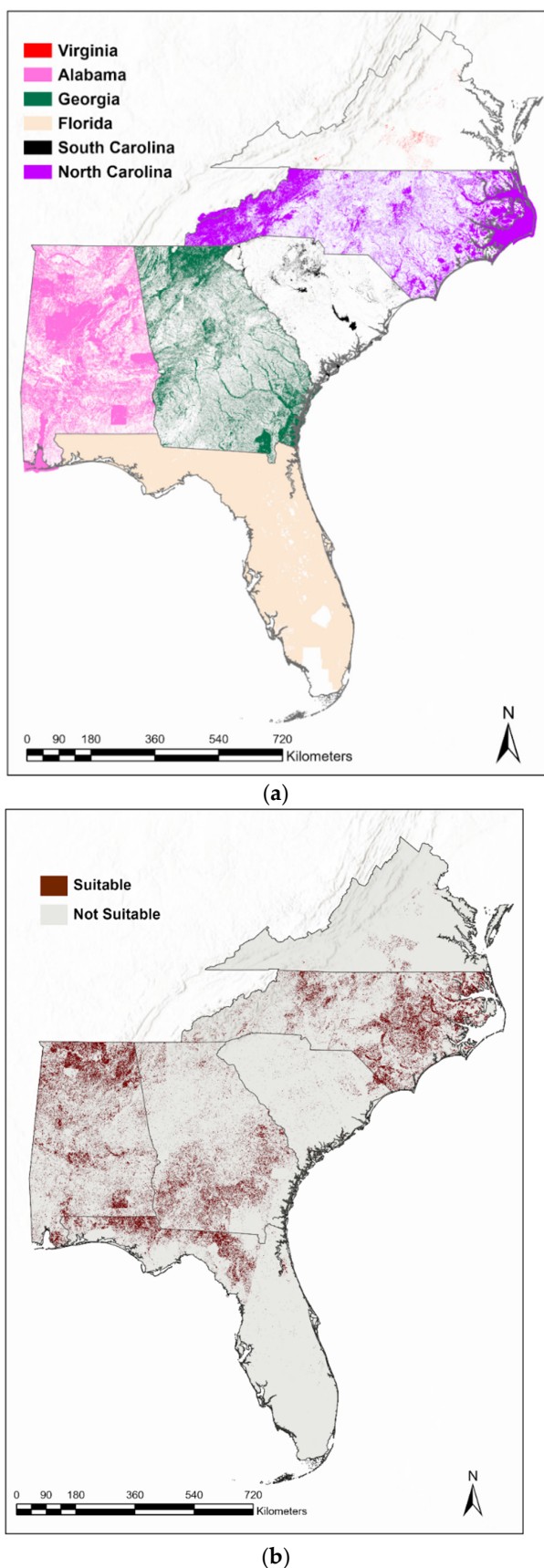

(**a**)

(**b**)

**Figure 4.** *Cont.*

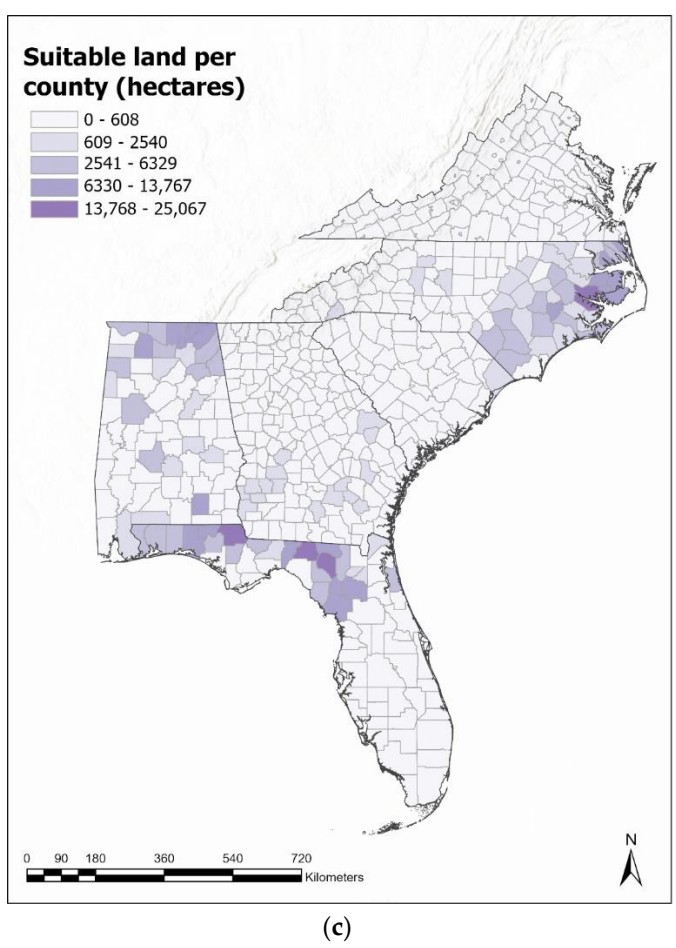

(**c**)

**Figure 4.** (**a**) Marginal lands (Classes 5–8) of six southeastern United States; Alabama, Florida, Georgia, North Carolina, South Carolina, and Virginia derived from the USDA's SSURGO (NRCS, 2015). (**b**) Potential suitable sites for establishing SRWC in six southeastern United States; Alabama, Florida, Georgia, North Carolina, South Carolina, and Virginia. The marginal land class (5–8), the corn, soybean, and tobacco crop data layers, and the non-federal lands data layers were pairwise intersected and clipped to boundary layers. (**c**) Potential suitable sites for establishing SRWC in six southeastern United States per county in Alabama, Florida, Georgia, North Carolina, South Carolina, and Virginia. The marginal land class (5–8), the corn, soybean, and tobacco crop data layers, and the non-federal lands data layers were pairwise intersected and clipped to boundary layers for result.

We combined the marginal land classes and the cropland data layer (CDL) [206] to identify marginal land under row crops (corn, soybean, and tobacco crops) in the states that could potentially be converted to SRWC. The CDL is a 30 m georeferenced raster data layer created annually for the continental United States using moderate resolution satellite imagery. Corn, soybeans, and tobacco were identified as the most cultivated row crops in these southeastern states [207]. The results of combined marginal classes and cropland data layer show a general reduction in land size for all states: North Carolina has the largest land area at ~185,276 hectares, while Virginia has the smallest at ~1125 hectares (Table 3). When the non-federal lands data layer was included in the analysis, we found AL, FL, GA, and NC have more suitable sites for the establishment of SRWC (Figure 4b). Further, we analyzed the land suitability by county (Figure 4c). The results show that NC, FL, and AL have more counties with suitable lands, ranging from approximately 2541 hectares to 25,067 hectares where SRWCs could be established. The counties in GA that have suitable lands ranged from approximately 688 hectares to 2540 hectares.

**Table 3.** Marginal class lands under corn, soybean, and tobacco in six Southeastern states.

| States | Marginal Class Lands (5–8) (Hectares) | Marginal Class Lands Under Corn, Soybean, & Tobacco (Hectares) |
| --- | --- | --- |
| Alabama | 7,743,297 | 96,414 |
| Florida | 48,474,354 | 161,939 |
| Georgia | 7,512,479 | 28,831 |
| North Carolina | 3,102,119 | 185,276 |
| South Carolina | 613,394 | 2373 |
| Virginia | 115,585 | 1125 |

Marginal land class was analyzed using the USDA's SSURGO land capability data and the National Agricultural Statistics Service cropland data layer—CDL.

The use of remote sensing and geospatial analysis in mapping potential suitable sites for SRWC establishment in the Southeast, offers an opportunity for farmers and landowners to establish and manage SRWC with greater efficiency. However, given the high cost of the commercial satellite imagery, an emerging field of remote sensing that relies on unmanned aircraft systems (UAS) is now commonly used to monitor, assess, and manage agroforestry activities [208]. The UAS can offer enhanced spectral and spatial resolution, operational flexibility, and affordability to the user, and has been successfully used for studies such as crop vigor assessment [209], biomass productivity estimation [210], disease monitoring [211], forestry operations [212], wildfire detection [213], and forest preservation [214]. Thereby offering stakeholders an accurate decision-support tool for land management in the near future.

## 7. Conclusions and Recommendations

From our findings, SRWC provide wood products, fiber, bioenergy feedstocks and ecosystem services while restoring and conserving the biodiversity of landscapes that have been degraded due to intensive farming and inadequate use of resource inputs (fertilizers, pesticides, and insecticide). Results have indicated the potential for the following ecosystem services with SRWC on degraded agricultural lands: (i) above and below ground carbon sequestration; (ii) increased biodiversity; (iii) restoration of soil health; (iv) reduced nitrogen and sediment loss; and (v) improved water quality. However, more experimental and field data would be valuable for the validation of models to accurately estimate biomass productivity and associated ecosystem services at regional and global scales. Further, low silvicultural inputs (fertilization, irrigation, pesticides, and insecticides) should be employed during establishment and maintenance of the SRWC plantation to increase optimal productivity and economic viability.

SRWC as purpose-grown feedstocks for biomass production is considered a low marginal-value commodity. Hence, there is a need for comprehensive cost–benefit analysis to optimize the economic and social benefits of SRWC. This includes valuing the accompanying ecosystem services that SRWC provide as well as incentives that encourage the production of SRWC on degraded agricultural lands. This is particularly important for the growing voluntary carbon markets. In addition, it is necessary to assess the tradeoff between the production costs of SRWC on degraded agricultural lands and the consequences for ecosystem services. In the southeastern U.S., where SRWC have more advantage for biomass production compared to many regions, increase in market price and diversified use of SRWC feedstocks will enhance the resource-use efficiency of the integrated SRWC-agriculture systems, while encouraging adoption among farmers and landowners. Furthermore, plantation establishment must be maximized for environmental performance and operational efficiency to be economically viable. For instance, farmers should be able to establish SRWC plantations with the same efficiency and probability of success as they achieve with other crops.

From the SRWC site suitability maps, the outlook for biomass production from SRWC in Alabama, Florida, Georgia, and North Carolina looks promising. Though the county level site suitability provides useful information, the potential combination of SRWC

carbon sequestration rates per county area will provide more insights on the increase of farm C storage due to the incorporation of SRWC into conventional agriculture, thereby highlighting the bioenergy potential in the US Southeast. The output can be used as a communication tool with stakeholders, landowners, woody biomass industries, and lawmakers. The use of remotely sensed data and geospatial analysis provide insight and geospatial information on site suitability, thereby, minimizing potential negative environmental impacts and the "food vs. fuel" debate in terms of land availability, while maximizing environmental sustainability.

In conclusion, farmer interests and perceptions are important when developing new farming strategies to ensure the success and sustainability of a new farming system. This is because farmers are likely to engage in a new farming system, such as growing woody bioenergy crops, when there are established markets and associated environmental benefits. As a result, the interrelationship between economics, ecosystem services, environmental sustainability, and societal development can be more successful through collaborative management strategies that incorporate results from research, and a feedback/information loop between stakeholders. Therefore, environmental and agricultural policies should focus on conserving biodiversity on former agricultural landscapes and increasing crop productivity in a sustainable manner. Innovative policies can be created through on-farm evaluation, actor diagramming, and prototyping sessions that encourage farmer attitudes towards growing SRWC, for the restoration of degraded agricultural lands and the advancement of nature-based solutions to environmental challenges.

**Author Contributions:** Conceptualization, O.J.I.; Methodology, O.J.I., H.M., S.S. and S.B.; Formal Analysis, O.J.I. and M.A.; Investigation, O.J.I., H.M., S.S. and S.B.; Data Curation, O.J.I., H.M., S.S. and S.B.; Writing—Original Draft Preparation, O.J.I., H.M., S.S. and S.B.; Writing—Review and Editing, O.J.I., M.A., H.D.R.C., J.I., J.B., J.L.H. and J.S.K.; Visualization, O.J.I. and M.A.; Supervision, J.B., J.LH. and J.S.K.; Project Administration, J.LH and J.S.K. All authors have read and agreed to the published version of the manuscript.

**Funding:** This research received no external funding.

**Data Availability Statement:** Not Applicable.

**Conflicts of Interest:** The authors declare no conflict of interest.

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
