# Peer review of "Integrating Short Rotation Woody Crops into Conventional Agricultural Practices in the Southeastern United States: A Review"

_land, doi:10.3390/land12010010_

Round 1

Reviewer 1 Report

In this review article, the authors utilize a synthesis focused approach to analyze the viability of short rotation woody crop (SRWC) viability in the US Southeast. The stated purpose of this article is to “provide insights that influence policy development, impacting widespread adoption of integrated SRWC energy farming in the southeastern U.S.” and they do this through the analysis of hundreds of sources from across the US and European related to bioenergy development trends. They also investigate the benefits of SRWC in degraded agricultural lands as well as its application in row crop agriculture. They also investigate Remote Sensing and GPS technologies related to SRWC implementation targeting and management.  articles also began with a larger contextualization of the SRWC topic within the UN’s Sustainable Development Goals (Goals 3, 7, 13, 15).

The review and distillation of the sources is very well organized and readable. The overall literature review on the various aspects of the topic is thorough. The addition of the GIS land suitability maps (figure 4 ff) based on the SSURGO and Cropland Data Layer sets provides a good visual supplement to analysis. The analysis is also strong in the context of situating SRWC in the context of land reclamation, the emerging carbon markets, as well as stating some limiting factors/controversies (i.e., food or fuel debate) and issues related to economies of scale (i.e., the CARB carbon market vs voluntary carbon market). These discussions make for a well-rounded and significant review of the literature on SRWC.

There are a few aspects of the article that need to be expanded, however. First, the article starts off by situating the SRWC into the larger UNSDGs. This interesting thread was abandoned immediately after mentioning it. It would be good to weave the applicable SDGs into each section throughout the article. It allows the article to explore the broader global and sustainable aspects of the SRWC technique and takes it beyond the US agricultural sector economic and mechanical focus of the article (though this it is arguably a US focused article). The lack of return to the SDGs makes it seem like the idea was forced in by a reviewer rather than a necessary aspect of the article.

I would also like to see more analysis/integration of the USDA BCAP into the discussion as well as an expansion of the voluntary carbon market section. I feel that more is needed on how farmers can enroll in voluntary carbon market, a discussion of payment amounts, an analysis of the solvency and permanence of the program, as well as a discussion of any bureaucratic hurdles. Since this is the most viable route for SRWC farmers, it seems critical to the future implementation and expansion of SRWC. Finally, one major feature of the article that was actually downplayed in the end was the statement in ll 118 and 119 “Further, this review may provide insights that influence policy development, impacting widespread adoption of integrated SRWC energy farming in the southeastern U.S.” Since this is not a policy review or development article, it is not a major expectation; however, a more policy oriented conclusion could be derived from the various agencies and applications discussed. If no policy framework will be discussed, it might be argued that the statement quoted above should be removed.

Reviewer 2 Report

Dear authors

Thank you for giving me the opportunity to read your work. First, I am sorry for taking so long to review it. The manuscript “Integrating Short Rotation Woody Crops (SRWC) into Conventional Agricultural Practices in the Southeastern United States:  A review” is very interesting. The study done by authors is a relevant topic in research and denotes a lot of work. Congrats on that. References are adequate and up-to-date. Figures have poor quality, specially the maps.

Nevertheless I have some questions/doubts and suggestions. Mainly, I just want to start some discussion on specific points and clear some points. I have some doubts about the methodology. First authors said they used “Google Scholar, Research Gate, CAB Directs, Web of Science, AGRIS, and university catalogs” in their search. However, why leave one of the two major sources (scopus) out and use research gate that is full of grey literature (we can upload whatever we want)?

Second, to me “short rotation woody crops in southeastern United States, agroforestry in southeastern United States, integrating bioenergy trees and row crops, integrating bioenergy trees and agriculture, emerging technologies in agroforestry, remote sensing in agroforestry, GIS in agroforestry” are no keywords. In some cases are complete sentences. For shure, this was not the slection structure used for searching for articles. Perhaps something like a SQL statement used is scopus and WoS?

Third, there is no information of the protocole used for the literature review. Did the author followed the PRISMA (the worldwide recommended method nowadays) or another one?

Finally, when authors state that “Due to the  limited amount of research that has been done on integrating SRWC into conventional agriculture systems in the southeastern United States, we included studies from other regions of the country and Europe to buttress the discussion”, then the title makes no sense any more since this literature review is nome more about “Integrating Short Rotation Woody Crops (SRWC) into Conventional Agricultural Practices in the Southeastern United States”.

Regarding line 165 do authors considered the use o Eucalyptus (Eucalyptus spp) and specially clones, a way to achieve sustainable development?

In lines 491 to 501 authors state tha farmers are predisposed to adopt SRWC but in exange of financial support. Do you think this is the way and that this traces a sustainable path?

In line 548 authors say “We analyzed the land suitability of SRWC” and the following text is all of data analysis performed by authors. However, the work intend to be a review was not it? Now I do not know what the right perspective to analyse this work and once again, the question of the title adequability comes forward.

As minor comments, I would highlight:

Line 34 – please replace “The UN Sustainable Development Goals” by The United Nations (UN) Sustainable Development Goals (SGD)”

Line 60 – “restoring degraded soils to supply biomass for energy can ensure access to affordable, reliable, sustainable and modern energy”. This is explained some settences forward in the text but in my oppinion the explanation should come right after the statement.

Line 69 – please define “AL, AR, FL, GA, LA, MS, NC, SC, TN, VA”, worldwide readers do not have to know the akonymus of the USA states.

Line 70 – please replace “U.S. net farm” by United States (U.S.) net farm

Line 77 – please replace “the southeastern U.S. (S.E.)” by the southeastern (S.E) U.S.

Line 98 please replace southeast by SE

Line 124 – It would be useful to have a figure with the study area and the identification of the states.

Line 131 – please replace GIS by Geographic Information Systems (GIS)

Line 223 – In line 44 authors used “tons/hectare/yr“ and now “Mg ha−1”. Please use always the same type of representation.

Line 230 – Table 1 is not mentioned in the text.

Line 298 to 311 – the leterring is diferent from the remaining text

Line 337 – please provide a space between 57mm

Line 557 – In my oppinion figure 4a is unecessary since we have table 3

Line 583 – the legend title of figure 4c cannot be Summurized Area in HECTARES. I also think the caption of the figure is wrong. It looks the same than figure 4b.

Sorry for being so picky,

Best regards and continue the good work

Reviewer 3 Report

The article is about the benefits of integrating Short Rotation Woody Crops (SRWCs) into conventional agriculture, with reference to the southeastern States of USA. The Authors consider environmental benefits such as improved soil fertility and increased biodiversity, weed reduction, and increased carbon sequestration, integrating the discussion with economic analyses (from energy production to carbon credits), reports on landowners’ perception and the use of remote sensing and geospatial analysis for identifying the most suitable areas for SRWCs. The article is well written, based on a robust data consultation (testified by 231 references), and, despite its length, pleasant to read. As such, it is certainly suitable for publication.

I’d have just some comments and questions to broaden the discussion (but I don’t know if they fall outside the topics of study of the consulted articles)….

-SRWCs are surely more effective and interesting in marginal agricultural lands and, as the authors note, there is still “a need for comprehensive cost–benefit analysis to optimize [their] economic and social benefits”. I believe that in more productive contexts, the introduction of SRWCs may involve economic losses - even unsustainable, especially for small farmers - deriving from harvest losses (during the 3-5 years of rotation) and the costs for labor and plant material purchase and transport. Is there any study addressing this issue?

-Is the employment of SRWCs suitable/advisable even in tree orchards (as they are part of conventional agriculture)? The effect on carbon sequestration is obviously less prominent when compared to agronomic tree species such as peach or olive, which, although not of primary importance in the considered states, have -however- a certain diffusion. It might be interesting to differentiate between these aspects.

-Is biomass combustion (and consequent CO2eq emissions) considered in the carbon sequestration analyses you reported?

Whereas SRWCs may increase biodiversity, might -conversely- the cutting of these trees after 3-5 years be a danger for avifauna?

-Super- minor observations

-line 69, you listed 10 southern states, please use also the names in full (for the first time)

-When you quote the references in the text you can avoid to insert the year of publication before the square brackets (e.g. Griffiths et al., 2019 ; Fischer et al., 2017  or Joshi and Mehmood (2011))

-lines 298-311 You used a different font

Round 2

Reviewer 4 Report

The authors have made significant improvements in the article.